# Foodborne Pathogen Dynamics in Meat and Meat Analogues Analysed Using Traditional Microbiology and Metagenomic Sequencing

**DOI:** 10.3390/antibiotics13010016

**Published:** 2023-12-21

**Authors:** Francesco Bonaldo, Baptiste Jacques Philippe Avot, Alessandra De Cesare, Frank M. Aarestrup, Saria Otani

**Affiliations:** 1Department of Food and Drug, University of Parma, 43124 Parma, Italy; francesco.bonaldo2@studio.unibo.it; 2Department of Veterinary Medical Sciences, University of Bologna, 40064 Ozzano Emilia, Italy; alessandra.decesare@unibo.it; 3Research Group for Genomic Epidemiology, National Food Institute, Technical University of Denmark, 2800 Lyngby, Denmark; baptiste.avot@geneviatechnologies.com (B.J.P.A.); fmaa@food.dtu.dk (F.M.A.)

**Keywords:** foodborne pathogens, meat, meat analogue, plant-based meat, metagenomics

## Abstract

Meat analogues play an increasing role in meeting global nutritional needs. However, while it is well known that meat possesses inherent characteristics that create favourable conditions for the growth of various pathogenic bacteria, much less is known about meat analogues. This study aimed to compare the growth and survival of *Escherichia coli* HEHA16, *Listeria monocytogenes*, *Salmonella enterica* Typhi, *Cronobacter sakazakii*, and a cocktail of these bacteria in sterile juices from minced chicken, pig, and beef, as well as pea-based and soy-based minced meat. Traditional microbiology and next-generation sequencing of those metagenomes were employed to analyse the pathogen variability, abundance, and survival after an incubation period. Our findings show that all the meat juices provided favourable conditions for the growth and proliferation of the studied bacteria, with the exception of *E. coli* HEHA16, which showed lower survival rates in the chicken matrix. Meat analogue juice mainly supported *L. monocytogenes* survival, with *C. sakazakii* survival supported to a lesser extent. A correlation was observed between the traditional culturing and metagenomic analysis results, suggesting that further work is needed to compare these technologies in foodborne setups. Our results indicate that plant-based meats could serve as vectors for the transmission of certain, but likely not all, foodborne pathogens, using two accurate detection methods. This warrants the need for additional research to better understand and characterise their safety implications, including their potential association with additional pathogens.

## 1. Introduction

Foodborne infections are a growing global public health threat, and 600 million cases of foodborne illnesses are reported annually [1]. These illnesses predominantly stem from foodborne pathogens, which account for approximately 70% of all foodborne diseases [2]. The presence of foodborne pathogens not only poses a risk to human health but also has adverse effects on both individuals and the overall economy, given their detrimental health impacts [3].

The pathogens that currently account for the highest number of outbreaks related to food are pathogenic *Escherichia coli*, *Salmonella* spp., *Campylobacter* spp., *Listeria monocytogenes*, and *Staphylococcus aureus* [4]. In addition, other microorganisms such as the opportunistic pathogen *Cronobacter sakazakii* represent an emerging threat considering their ability to contaminate a wide range of products and their capacity to cause a variety of diseases with a mortality rate of 40–80% in infants and immunocompromised people [5]. Therefore, it is crucial to consider the potential transmission of these pathogens when evaluating the safety of meat and meat analogues, which are often involved in episodes of foodborne illnesses [6].

Meat is considered a nutritious food due to its composition of essential nutrients, particularly high-quality proteins, as well as a diverse range of fatty acids and micronutrients. However, the projected increase in the global population to 9 billion by 2050, along with the imperative to produce food more sustainably, has prompted the development of alternative protein sources, such as meat analogues [7]. These alternatives are predominantly made from easily accessible and cost-effective protein sources like soy and peas [8]. The production process, which usually includes an extrusion phase, leads to products with a better microbiological profile than meat products [8]. However, these products have characteristics such as a neutral pH, high protein content, and high moisture content [9], which make them highly susceptible to microbial growth. For example, tempeh products were associated with an outbreak of *Salmonella enterica* Paratyphy B in 2012, which resulted in 89 cases of illness [10]. Consequently, it is crucial to understand the survival and growth mechanisms of pathogenic bacteria in meat analogues to implement preventive strategies and safeguard their safety.

Pathogenic bacteria have traditionally been studied using traditional microbiology (culturomics). The use of next-generation sequencing (NGS) is becoming increasingly popular in food microbiology research because it allows for a comprehensive analysis of the food microbiome that goes beyond simple pathogen detection, and it allows for the characterisation of communities (e.g., pathobiomes) [11,12,13,14]. This advanced technology enables the simultaneous detection of multiple pathogens and the identification of various pathovars of the same pathogen, as well as the determination of specific genes of interest [15] and which co-occurring bacterial taxa might play a role in pathogen persistence or disappearance.

Meat fluid and juices have previously proven valuable for laboratory-based evaluations of bacterial pathogen survival and persistence that could mimic real-world conditions, and they can be an important source of bacterial contamination on food processing surfaces [16,17]. Chicken juice can, for example, modulate transcriptional responses in pathogenic bacteria, such as in the case of *Campylobacter jejuni*, where an increase in cell survival has been identified [18].

In this study, we compared meat and meat analogues for their ability to promote known foodborne pathogen growth and survival. To evaluate foodborne pathogen occurrences, we combined culturomics with deep metagenomic next-generation sequencing of meat and meat analogues to accurately assess the bacterial pathogen growth and survival in different meat types.

## 2. Material and Methods

### 2.1. Sample Collection and Preparation

Five different meat samples were used to evaluate pathogen growth and survival: minced chicken meat, minced beef, minced pork, a minced meat substitute made mainly from peas, and a minced meat substitute made mainly from soy. All meat products were purchased from Danish supermarkets (Lyngby, Denmark). The experiment was carried out in triplicate for each meat type.

We used meat and meat analogue juices as representative of the meat type to mimic real-life contamination and study the effect of matrix promotor factors on pathogen growth and survival rather than the matrix itself. Meat and meat analogue juices were obtained by mixing 50 g of the sample with 65 mL sterile PBS (phosphate-buffered saline) and blending the mixture for complete homogenisation. To remove large particles, the homogenised mixture was centrifuged for 10 min at 5000× *g*, and the supernatant was collected. In the case of the soy plant-based minced meat, the procedure was the same, except that 180 mL of PBS was added to the 50 g of sample due to the sample’s dry nature. All supernatants collected from the meat and meat analogue juices were double-filter-sterilised. Initially, a 0.16 µm filter was employed to remove large debris, followed by the use of two 0.02 µm filters for bacterial removal. The sterile meat and meat analogue juices were used immediately after preparation to prevent any changes in the matrix chemistry or sterility. Sterile meat and meat analogue juices were used as negative controls in the pathogen survival assays.

### 2.2. Bacterial Pathogen Preparation

Four pathogenic bacteria, namely, *Salmonella enterica* Typhi 2017 F3, *Escherichia coli* HEHA16, *Cronobacter sakazakii* ATCC 10250, and *Listeria monocytogenes* HH-E-1, were selected from the National Food institute, Technical University of Denmark, surveillance collection. The bacteria were revitalised from the −80 °C collection via double cultivation on LB agar (Merck KGaA, Darmstadt, Germany), which was then incubated at 37 °C for 24 h. One colony from each isolate was taken and transferred to BPW (buffer peptone water) (Thermo Fisher Scientific, Waltham, MA, USA) medium. After incubation at 37 °C for 24 h, the OD was measured to determine the cfu/mL (cfu: colony forming unit) (Appendix A). The concentration values were calculated by referring to (Appendix A) those derived from González-Pérez et al. (2019) [19]. *Cronobacter sakazakii* cell density was calculated by taking *Escherichia coli* 0157:H7 as the reference, while the cell density of the remaining taxa was measured using the OD measurements. To spike the sample with a pathogen community, a cocktail comprising equal amounts of bacterial cells from each species was also prepared (Appendix A).

### 2.3. Bacterial Inoculation and Culturomics

The activity and growth patterns of indigenous and artificially introduced spoilage/pathogenic bacteria were observed in both meat and meat analogues. Sterile juice samples from each matrix were spiked with each of the bacterial pathogens, as well as the cocktail mixture (Figure 1). The spiked meat and meat analogue juices were incubated for 3 consecutive days at a bacteria-favoured temperature (37 °C). To assess the bacterial survival rate, each matrix with a spiked-in bacterium was cultured on agar plates for bacterial cell counting on each day (day 1, day 2, and day 3) of the matrix/bacteria incubation. *Salmonella enterica* Typhi was plated on XLD agar (Thermo Fisher Scientific), while the other microorganisms were plated on LB agar (Merck, Germany). Due to the high bacterial density, a series of dilutions were used to enable CFU counting. The matrix/bacteria were diluted to 10^−6^ and incubated at 37 °C for 24 h, and bacterial CFUs were counted manually on light-illuminated Petri plates. After colony counting, the CFU numbers were corrected with the dilution factors to present the accurate cfu numbers without dilutions. For each meat and meat analogue matrix, a control of only sterile matrix juice without bacterial inoculation was included and plated on LB agar (Merck, Germany) on each day (day 0, day 1, day 2, day 3) after incubation at 37 °C.

### 2.4. DNA Extraction and Metagenomic Sequencing

Next-generation sequencing of metagenomes was also carried out, alongside culturomics, on each spiked matrix with each bacterial species and the cocktail on each day that the matrices were incubated, including day 0 (when the bacteria were inoculated and before starting the incubation). DNA was extracted from 5 mL of the homogenate of meat and meat analogue juice with the spiked-in bacterium via centrifugation for 10 min at 5000× *g*. The pellet was then stored at −20 °C until extraction. DNA extraction was performed using DNeasy Blood and Tissue (Qiagen, Venlo, The Netherlands) following the manufacturer’s instructions with the following modification: approximately 0.2 g of pellets was mixed with 180 µL of ATL buffer (Qiagen). After vortexing for a few seconds, 20 µL of proteinase k was added and mixed. Then, 200 µL of AL buffer (Qiagen) was added and mixed by vortexing. Afterwards, the samples were incubated for at least one hour at 56 °C, followed by adding 200 µL of 100% ethanol and mixing. Then, 600 µL of the lysate was used for DNA isolation using the provided columns. After cleaning, the DNA was eluted in 50 µL of AE buffer (Qiagen) at 56 °C. The extracted DNA was quantified using a Quanit4 Fluorometer (Thermo Fisher Scientific). Libraries for metagenomic sequencing were prepared from the DNA using a KAPA HyperPlus PCR-free kit (Roche) following the manufacturer’s instructions. DNA samples were sequenced on a Novaseq6000 platform (Illumina, San Diego, CA, USA). All the samples were sequenced in paired-end mode with a sequencing depth of 85 million reads minimum per sample.

Blank controls of sterilised distilled water and sterilised meat juice were included. The DNA concentrations in those were undetectable, and, hence, sequencing libraries from the controls failed due to undetectable DNA yields.

### 2.5. Bioinformatic Analyses

Raw read quality was checked using FastQC (FastQC version 0.11.5, https://www.bioinformatics.babraham.ac.uk/projects/fastqc/, accessed on 15 June 2022), and the trimming of raw reads was performed with bbduk2 (part of BBmap v36.49). Sequence quality was evaluated with FastQC v0.11.5 before and after quality processing. KMA 1.3.27 [20] was used to align the reads with the bacterial database. The reads were assigned to bacterial taxa using a bacterial database that contains complete and partial bacterial genomes that were manually curated at Danmarks Tekniske Universitet (DTU) and to another internal database of only the targeted bacterial species (*Salmonella enterica* Typhi, *Listeria monocytogenes*, *Escherichia coli* HEHA16, and *Cronobacter sakazakii*). A visual presentation of the output was developed in R [21].

## 3. Results

### 3.1. Viability of Bacterial Pathogens in Various Meat and Meat Analogues Using Culture-Dependent Approach

To assess the bacterial pathogens’ abilities to grow in the tested meat and meat analogues, all counted bacterial colonies on days 1, 2, and 3 were adjusted for the initial inoculum present in the respective matrices. By correcting for the initial inoculum, a clearer representation of the bacterial pathogen’s ability to proliferate within the tested matrices was obtained. Sterile matrix juices were also cultured on LB agar (Merck, Germany) plates on all days to confirm the effective removal of culturable microorganisms following double-filter sterilisation.

The plate count results showed that chicken juice did not support favourable growth for *Escherichia coli* HEHA16. This was evident from the reduction in the bacterial concentration observed on days 1, 2, and 3, compared to the inoculum number pre-spiking (Figure 2; Appendix A; Appendix A). In contrast, the other two animal matrices exhibited higher abilities for *E. coli* bacterial growth than the meat analogue juices, as evidenced by consistent growth between days 1 and 2 of the experiment. This was particularly present in the beef meat matrix, where *E. coli* colonies were increasingly prevalent as the incubation continued. Regarding *Listeria monocytogenes*, plate counts revealed that this bacterium thrived best in plant-based meat analogues compared to meat, specifically in soy juice, where the bacterial concentrations remained relatively stable over the three-day experiment (Figure 2; Appendix A). However, *L. monocytogenes* colonies decreased in the meat matrices (Figure 2; Appendix A). In contrast, *Salmonella enterica* Typhi exhibited better proliferation in the animal matrices, particularly in chicken juice, where the average number of colonies recorded on day 1 was twice that of beef and nearly double that of pork (Figure 2; Appendix A). In terms of bacterial survival, pea juice showed the highest stability with consistent concentrations over the three-day period when infected with *S*. *enterica* Typhi. *Cronobacter sakazakii* showed the highest concentrations in the animal matrices, with a consistent increase between days 1 and 2. However, all *C. sakazakii* colonies decreased as the incubation continued (Figure 2; Appendix A). All control cultures (sterile juice meat and meat analogues) remained sterile without any visible CFUs during or after 3 days of incubation.

### 3.2. In-Depth Detection of Bacterial Pathogens in Various Meat and Meat Analogues Using Metagenomic Sequencing Approach

The DNA yields from our 60 spiked-in meat and meat analogues ranged between 4.3 and 640 ng/µL, from pea-based and pig meat inoculated with *Listeria* and *Salmonella*, respectively.

All 60 metagenomic samples were sequenced and resulted in 6.3 billion high-quality reads, ranging between 75 and 142 million reads between the metagenomes. Among the samples spiked with the bacterial cocktail, 99% of the identified bacterial taxa from these metagenomes belonged to the four bacterial species inoculated. In the case of meat and meat analogues where a single bacterium was introduced, almost the entire sample output consisted solely of that specific bacterial species, with minimal variations in the relative abundances of the species observed across the incubation days (days 1, 2, and 3) (Appendix A). As a result, for all subsequent metagenomic analyses, we focused on the matrix/bacterium samples where a cocktail of the four bacteria was spiked (Figure 1), as these samples exhibited negligible variations attributable to the different bacterial species. 

The metagenomic sequencing of the bacterial cocktail in the different matrices showed that *Escherichia coli* HEHA16 grew the best in chicken juice only for one day, and its abundance declined rapidly over days 2 and 3 (Figure 3 and Appendix A). *E. coli* HEHA16 did not show great growth in the remaining matrices. However, it survived in the pig, soy-based, and pea-based matrices over the three days of incubation, unlike in the chicken- and beef-based ones. In the case of *Listeria monocytogenes*, the values obtained were consistently low across all samples, showing a decreasing trend throughout the experiment (Figure 3 and Appendix A). However, pea juice showed the highest survival rate for *L. monocytogenes* compared to the remaining three bacteria, with a minimal decrease in the relative abundance from 2.4 on day 1 to 1.9 on day 3 (Figure 3 and Appendix A). Regarding the matrices spiked with *Salmonella enterica* Typhi, it was apparent that animal-based meat juices favoured *S. enterica* Typhi growth compared to plant-based ones (Figure 3 and Appendix A). *Cronobacter sakazakii* was present in higher relative abundances in the meat analogues than in the animal-based ones, with approximately stable values across the three days of the experiment (Figure 3 and Appendix A).

## 4. Discussion

In particular, soy, pea, and wheat protein have emerged as the primary ingredients used in the production of numerous mainstream plant-based meat analogues [22]. There is still very little research that evaluates how these meat analogues promote or prevent the growth of traditional foodborne bacterial pathogens, as well as other potential novel pathogens. Considering their neutral pH, high protein content, and high moisture levels [9], these meat analogues might exhibit susceptibility to microbial proliferation similar to that of conventional ground beef.

Here, we undertook a comprehensive examination of the microbial quality and safety of meat products, including plant-based meat analogues, using not only traditional culturing methodologies but also a comprehensive metagenomic approach. We used sterile meat juice to study the effect of meat and meat analogue promotor factors on pathogen survival rather than the matrix itself or its innate bacterial community. We observed and assessed the changes in the artificially added bacterial pathogens in plant-based meats, as well as in animal-based meats, and then we monitored the survival of the artificially induced spoilage with pathogenic bacteria over time [16,23].

Cultivation conditions in the meat and meat analogue juices were employed to mimic the realistic environment found in the meat processing industry. To create a representative model, minimally processed meat juice was utilised. We demonstrated the viability of using meat juice as a means to examine the impact of foodborne pathogens and their biofilms, and how the meat environment and growth factors influences food pathogen proliferation without the meat itself as a medium.

We also deliberately selected a temperature of 37 °C to study bacterial growth in meat to simulate a worst-case scenario, as 37 °C is the optimal growth temperature for many bacterial pathogens. While meat products are typically stored in refrigerated or frozen conditions, it is important to recognise potential instances of temperature changes during transportation, storage, or handling. Such lapses in the cold chain can expose foods to temperatures conducive to rapid bacterial growth. Our findings, therefore, provide insights into the maximum potential risks under such adverse conditions.

The results from both the quantitative plate counts and metagenomic sequencing analyses showed that *Escherichia coli* HEHA16 does not grow and that its viability is not sustained during the incubation period in chicken juice. This suggests that *E. coli* lacks the capacity to proliferate and maintain an active state within this specific meat juice. The observed high abundances in the metagenomes on day 1 (13.66%) can be attributed to the presence of DNA originating from the initial bacterial inoculum on day 0. This phenomenon consequently elucidates the subsequent absence of *E. coli* on days 2 and 3. The high abundances on day 1 are likely due to metagenomic principles, as metagenomic sequencing primarily detects the presence of bacterial DNA rather than determining the actual viability of bacteria. Consequently, the detectable DNA content persists, even in cases where bacterial cells may no longer be viable.

Although *E. coli* is a known poultry contaminant [24], this could suggest that chicken meat juice does not support *E. coli* growth and survival; therefore, perhaps the meat itself is required for such proliferation. *E. coli* showed better survival on cultures in all other meat and meat analogue juices, with beef being more favoured than soy and pea matrices, which is in agreement with previous reports [25]. This was also observed in our metagenomics data, except for beef, where not many reads were assigned to this *E. coli* strain. Considering that a different stress response was observed in different *E. coli* pathotypes, such as *Escherichia coli* 0157:H7 and *Escherichia coli* 0121, it is conceivable that different pathotypes and serovars could result in diverse growth and survival profiles in food [26]. *Listeria monocytogenes* showed better survival in the meat analogues using both methods, where pea juice was found to be the most supportive matrix in terms of bacterial survival. This might suggest that soy and pea matrices could provide favourable environments for the survival of *Listeria monocytogenes,* which might be similar to what others described with *L. monocytogenes* in other plant-based meat analogues [27]. They demonstrated that the bacterium was able to survive for up to 24 days in tofu stored at room temperature.

*Salmonella enterica* Typhi showed the best survival rates in all meat and meat analogue juices of all tested bacteria and using both detection methods. *S. enterica* Typhi, a distinct serotype primarily associated with humans, is linked to inadequate personal hygiene practices [28,29,30],. Among the different animal sources analysed, chicken juice has been identified as the most favourable matrix for *Salmonella*, yet it was the worst in terms of bacterium survival on cultures, as the CFUs decreased over time. All remaining matrices, including the meat analogues, were able to support *S. enterica* Typhi survival, as shown using both methods. 

*Cronobacter sakazakii* showed high survival rates over the incubation time in all meat juices, including in the plant matrices, using both methods. *C. sakazakii* is mostly known to contaminate dairy products (rehydrated milk) [31,32], and it is less likely to contaminate meat products. Our results also show that it has improved survival in plant-based meat analogues and could be more adapted to such matrices rather than animal-based ones.

Our study not only employs culture-dependent methods for foodborne pathogen detection but also combines next-generation metagenomic sequencing to detect pathogens that might not be detectable with traditional culturing. Metagenomics detects not only culturable bacterial taxa but also pathogens that we might not be able to culture, or that are not viable anymore, as shown in the *E. coli* example above. When comparing both methods’ abilities to detect the inoculated bacterial pathogens, a positive correlation was found between the two, confirming our results. However, the moderate correlation observed between the two detection methods is possibly attributed to the individual spiking of bacteria into the food matrices in the culturomic analysis, whereas metagenomic data were obtained from matrices spiked with a bacterial cocktail. As a result, it is conceivable that the other pathogens present in the matrix may have influenced the growth and survival profiles of the targeted bacteria of interest.

## 5. Conclusions

By using both of our comprehensive methods, we show that *S.* Typhi and *C. sakazakii* grew and survived over the incubation period in meat and meat analogue juices. However, *E. coli* and *L. monocytogenes* were more supported by different matrices; e.g., *L. monocytogenes* survived better in plant-based meat than in animal meat. This suggests that our evaluation for food safety is not only matrix-dependent but also pathogen-dependent. This is particularly important for the newly emerging plant-based meat products, where we show that three out of our four tested pathogens survived in all plant-based meat analogues. With the anticipated market growth for plant-based meats in the coming decade, the necessity of establishing a robust knowledge base concerning the microbial quality and safety of plant-based meat analogues is surfacing. Once introduced into plant-based meat analogues, both spoilage and pathogenic microorganisms have the capacity to not only survive but also proliferate. As such, while our findings give an upper bound of potential bacterial risks, they might not directly depict the risks under regular storage. Various factors, including storage temperatures, pathogen and spoilage microorganism types, the levels of native microorganisms, and the specific plant protein utilised, can all also influence the behaviour of different microorganisms. Building on our findings, there is a clear impetus for further research that examines bacterial growth in plant-based meat products under a range of temperatures, particularly those relevant to typical storage (refrigerated or frozen). Such studies would complement our worst-case scenario findings and provide a comprehensive risk assessment across various storage conditions.

Further studies investigating the relationship between specific food matrix nutrients and the growth and survival of the studied pathogenic bacteria would also complement the findings of this study, for example, a study evaluating how meat derivatives composed primarily of other protein sources, such as potato, wheat, and mung beans, may impact the growth and survival of pathogenic bacteria.

## Figures and Tables

**Figure 1 antibiotics-13-00016-f001:**
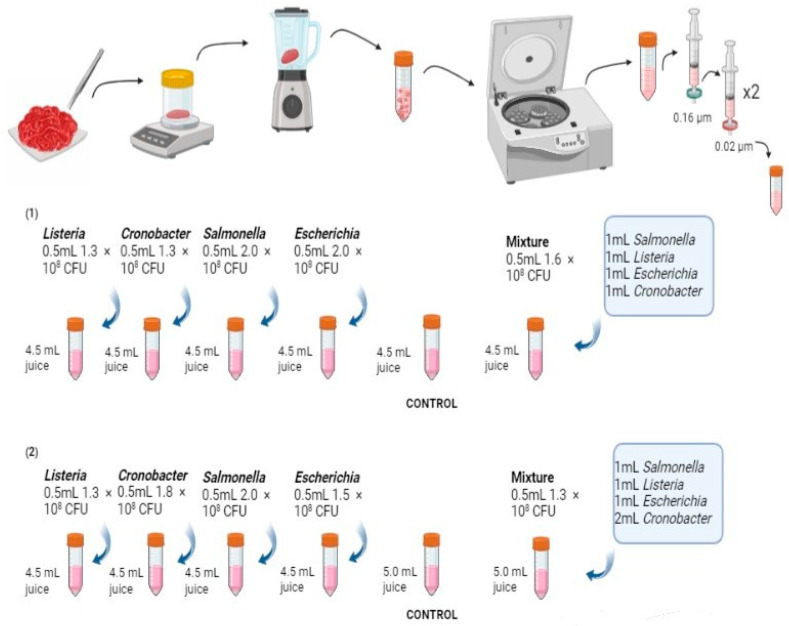
Experimental design and workflow. Row (**1**) represents the bacterial concentrations used to spike pig, pea, and soy juice. Row (**2**) represents the bacterial concentrations used to spike chicken and beef juice. Control tubes contain the sterile meat and meat analogue juice only.

**Figure 2 antibiotics-13-00016-f002:**
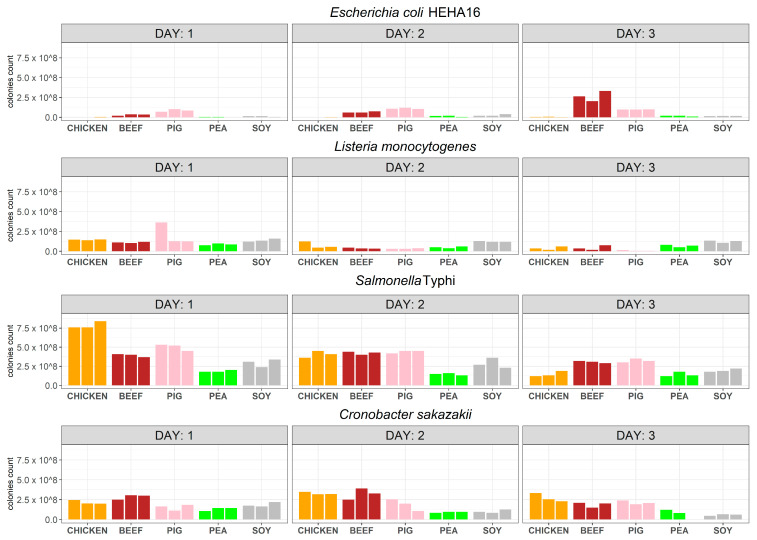
Foodborne bacterial pathogen growth and survival over time in four meat and meat analogues using culture-dependent approach. Each bar represents the counted CFUs after the initial artificial inoculation date (day 0). Each number was adjusted to represent only the bacteria growth after adding the starting bacterial inoculum, and all CFU counts were adjusted to CFU/100 µL volumes of the initial matrix/bacterial mix to be able to compare the outputs.

**Figure 3 antibiotics-13-00016-f003:**
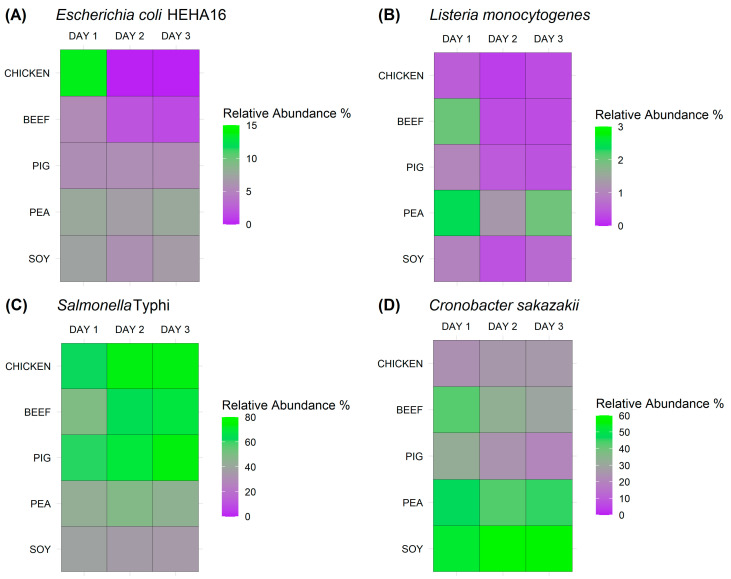
Foodborne bacterial pathogen growth and survival over time (as relative abundance values) collectively in the bacterial cocktail mixture. Growth and survival were measured in four meat and meat analogues using a culture-independent approach in metagenomics sequencing. Each heatmap represents the growth and survival of one bacterial taxon in the cocktail after the inoculation date (day 0), and each cell colour represents the relative abundance of the specific bacterium according to the colour scale provided.

## Data Availability

The entire metagenomic sequencing data can be publicly accessed through the European Nucleotide Archive (ENA) with the following project accession number: PRJEB63613 (Study ERP148765).

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
