# Peer review of "Foodborne Pathogen Dynamics in Meat and Meat Analogues Analysed Using Traditional Microbiology and Metagenomic Sequencing"

_antibiotics, 2023, doi:10.3390/antibiotics13010016_

Round 1

Reviewer 1 Report

Comments and Suggestions for Authors

The paper presents a major challenge, which has been widely discussed: Dynamics of pathogens in food analysed by traditional microbiology and metagenomic sequencing. The study has been conducted with appropriate methodology, it is well written, I only have two considerations:

1- Why is the R^2 in the figure 3 S so low?  I think it would be interesting to have a justification presented in the manuscript

2- In the discussion, I recommend a discussion on the issue of CFU and type of bacteria, given that some international food quality control legislation allows a limit on the presence of certain strains of bacteria.

Author Response

Reviewer 1:

The paper presents a major challenge, which has been widely discussed: Dynamics of pathogens in food analysed by traditional microbiology and metagenomic sequencing. The study has been conducted with appropriate methodology, it is well written, I only have two considerations:

1- Why is the R^2 in the figure 3 S so low?  I think it would be interesting to have a justification presented in the manuscript

“R2 is 0.33 when we looked at the correlation between culture-based methods and metagenomics. We agree with the reviewer that it is low. Which is why we did not emphasise on the correlation between metagenomics and culture-based methods since it is a moderate correlation. We added more information on this in lines: 336-341”

2- In the discussion, I recommend a discussion on the issue of CFU and type of bacteria, given that some international food quality control legislation allows a limit on the presence of certain strains of bacteria.

“We thank the reviewer for this point. However, due to the variations in international food quality control legislations, and as mentioned in the reviewer comment, we refrain from discussing these legislations. It is also outside the scope of the manuscript, as the manuscript is mostly on the differences between plant-based meat and meat ability to retain bacterial pathogens.”

Reviewer 2 Report

Comments and Suggestions for Authors

This study explored the growth and survival of Escherichia coli HEHA16, Listeria monocytogenes, Salmonella enterica Typhi, Cronobacter sakazakii, and a cocktail of these bacteria in sterile juices from minced chicken, pig, beef, as well as pea-based and soy-based minced meat using culture and metagenomic methods. The results showed matrix-dependent variations in bacterial pathogen growth and survival.

Some concerns:

1. Escherichia coli is diverse, and different pathotypes are assigned. Little is known about the E. coli HEHA16 used in this study.

2. Why use 0.16 μm filter followed by two 0.02 μm filters to sterilize?  In the figure 1, they were typo as 0.16 μg and 0.02 μg.

3. The results of single bacteria growth and survival is different from that of cocktail culture, especially for E. coli HEHA16 in the chicken juice at day 1. Though the authors discussed that this may be the viable bacterial cells. Are there other reasons? Is it possible to verify this using traditional culture?

Author Response

Reviewer 2:

This study explored the growth and survival of Escherichia coli HEHA16, Listeria monocytogenesSalmonella enterica Typhi, Cronobacter sakazakii, and a cocktail of these bacteria in sterile juices from minced chicken, pig, beef, as well as pea-based and soy-based minced meat using culture and metagenomic methods. The results showed matrix-dependent variations in bacterial pathogen growth and survival.

Some concerns:

  1. Escherichia coliis diverse, and different pathotypes are assigned. Little is known about the E. coli HEHA16 used in this study.

“We agree with the reviewer the E. coli is diverse with several pathotypes. This make selecting the right E. coli strains for such experiment complicated as the options are diverse. We chose E. coli HEHA16 as it is part of our E. coli collection that we studied several anti-gene characteristics in (e.g., Kjaergaard K, Schembri MA, Hasman H, Klemm P. Antigen 43 from Escherichia coli induces inter- and intraspecies cell aggregation and changes in colony morphology of Pseudomonas fluorescens. J Bacteriol. 2000 Sep;182(17):4789-96. doi: 10.1128/JB.182.17.4789-4796.2000. PMID: 10940019; PMCID: PMC111355. -- Kjaergaard K, Hasman H, Schembri MA, Klemm P. Antigen 43-mediated autotransporter display, a versatile bacterial cell surface presentation system. J Bacteriol. 2002 Aug;184(15):4197-204. doi: 10.1128/JB.184.15.4197-4204.2002. PMID: 12107137; PMCID: PMC135209. -- Klemm, P., Hjerrild, L., Gjermansen, M. and Schembri, M.A. (2004), Structure-function analysis of the self-recognizing Antigen 43 autotransporter protein from Escherichia coli. Molecular Microbiology, 51: 283-296. https://doi.org/10.1046/j.1365-2958.2003.03833.x) and we considered it a well studied K12 E. coli. (lines: 304-307)”

  1. Why use 0.16 μm filter followed by two 0.02 μm filters to sterilize?  In the figure 1, they were typo as 0.16 μg and 0.02 μg.

“Thank you for the reviewer for picking up on such small detail. We used first a 0.16 μm filters because the meat and meat analogue juice debris where too large for the smaller filter first (0.02 μm) and clogged it. Therefore, we ran the meat juice first through larger pores: 0.16 μm, to eliminate the large debris, then followed by smaller pore filters: 0.02 μm, to sterilise the juice. Those were added in lines: 109-111”

  1. The results of single bacteria growth and survival is different from that of cocktail culture, especially forE. coli HEHA16 in the chicken juice at day 1. Though the authors discussed that this may be the viable bacterial cells. Are there other reasons? Is it possible to verify this using traditional culture?

“We thank the reviewer for this comment. We added extra explanations in lines: 336-341”